# Oxygen–Ozone Therapy in Cervicobrachial Pain: A Real-Life Experience

**DOI:** 10.3390/jcm12010248

**Published:** 2022-12-29

**Authors:** Vincenzo Rania, Gianmarco Marcianò, Alessandro Casarella, Cristina Vocca, Caterina Palleria, Elena Calabria, Giuseppe Spaziano, Rita Citraro, Giovambattista De Sarro, Francesco Monea, Luca Gallelli

**Affiliations:** 1Operative Unit of Pharmacology and Pharmacovigilance, “Mater Domini” Hospital, Department of Health Science, University “Magna Graecia”, 88100 Catanzaro, Italy; 2Research Center FAS@UMG, Department of Health Science, University “Magna Graecia”, 88100 Catanzaro, Italy; 3Dentistry Unit, Department of Health Sciences, University “Magna Graecia”, 88100 Catanzaro, Italy; 4Department of Environmental Biological and Pharmaceutical Sciences and Technologies, University of Campania Luigi Vanvitelli, 81100 Caserta, Italy; 5Poliambulatorio Specialistico, Taurianova, 89029 Reggio Calabria, Italy; 6GalaScreen Laboratories, University of Calabria, Ed. Polifunzionale, Arcavacata di Rende, 87036 Rende, Italy; 7Medifarmagen SRL, University of Catanzaro and Mater Domini University Hospital, 88100 Catanzaro, Italy

**Keywords:** oxygen–ozone (O_2_–O_3_) therapy, intramuscular, cervicobrachial pain, efficacy, safety

## Abstract

This prospective, open-label clinical study was carried out to evaluate both the efficacy and safety of intramuscular paravertebral injections of an oxygen–ozone (O_2_–O_3_) mixture in patients with cervicobrachial pain. We enrolled 540 subjects affected by cervicobrachial pain referred to the Ozone Therapy Ambulatory at the Mater Domini Hospital of Catanzaro (Italy) and to the Center of Pain in Taurianova (Reggio Calabria, Italy). All the subjects (n = 540) completed the treatment and the follow-up visits. The subjects received a mean of 11 cervical intramuscular treatments with an O_2_–O_3_ mixture (5 mL) with an O_3_ concentration of 10 μg/mL bis a week. The improvement of pain was measured by a change in the mean of the Visual Analog Scale (VAS) score from baseline to the end of treatment and during follow-ups. Patient satisfaction was assessed at the end of treatment using the SF-36 Questionnaire. The development of adverse drug reactions was recorded. The mean (±standard deviation) VAS pain score at baseline, at the end of treatment, and during follow-ups showed a significant reduction in pain over time (*p* < 0.001). All the patients who were enrolled (n: 540) were pain-free after one year. According to the pain distribution, all subjects showed a significant reduction in pain over time in each group (*p* < 0.05). No significant differences were observed with respect to sex or age. No adverse events were observed during the study. In conclusion, we documented that the intramuscular injection of an O_2_–O_3_ mixture is an effective and safe treatment option for patients with cervicobrachial pain.

## 1. Introduction

Neck pain is a common and multifactorial disease, often associated with brachial pain [1]. Pharmacological and non-pharmacological therapies are commonly used to reduce pain and improve the quality of life [2]. Concerning new therapies, ozone (O_3_) could be an effective treatment option for musculoskeletal disorders and painful syndromes affecting muscles, tendons, and joints [3]. Ozone is a natural gas, first used in medicine for the antimicrobial treatment of wounds during the First World War, and nowadays, it is used in many areas of medicine [4]. Recently, several authors focused on the biological and therapeutic effects of oxygen–ozone (O_2_–O_3_) on inflammatory conditions and musculoskeletal diseases [5,6]. It has been reported that several proinflammatory mediators, e.g., prostaglandins (Pgs) and bradykinin, released by macrophages and leukocytes might be regulated by O_2_–O_3_ [7].

Moreover, O_2_–O_3_ decreases the production of reactive oxygen species (ROS) [8], and increases the release of nitric oxide (NO), reducing cell damage and improving tissue oxygenation [9,10].

In particular, the improvement of tissue oxygenation increases the use of glucose in cellular metabolism, protein metabolism, and erythrocyte activity. Taken together, it is easy to understand that O_2_–O_3_ has analgesic and anti-inflammatory activities also related to the increased stimulation of the secretion of both serotonin and endogenous opioids [11,12].

Moreover, other authors have suggested that after continuous administration O_3_ upregulates Nrf2, which reduces the intracellular signaling pathways of inflammation and increases antioxidant activity [13,14].

However, to date, the use of O_2_–O_3_ in the management of musculoskeletal or inflammatory pain has not been reported in the guidelines [15,16].

Therefore, in this study, we evaluated the efficacy and the safety of intramuscular paravertebral injections of an O_2_–O_3_ mixture in patients referred to medical care with cervicobrachial pain.

## 2. Materials and Methods

### 2.1. Study Design

We conducted an open-label prospective clinical study between 1 February 2021, and 31 August 2022, on patients with cervicobrachial pain referred to the Ambulatory of Pain Medicine of Mater Domini Hospital in Catanzaro (Calabria, Italy) and the Center of Pain Disease in Taurianova (Reggio Calabria, Italy). The study, approved by the Ethics Committee, was carried out according to the Good Clinical Practice guidelines and under the ethical principles of the Declaration of Helsinki. Before the beginning of the study, all participants signed a written informed consent form.

### 2.2. Experimental Protocol

At admission (T0), each patient, after medical history, underwent a physical examination, e.g., detection of vital parameters, thoracic and cardiac evaluation, abdominal palpation, and evaluation of osteotendinous and osteo-muscular activity. Particular attention was paid to the cervical brachial area to evaluate motility and the presence of points of greater pain. Radiological, ultrasound, and laboratory examinations were also employed. Pain severity was evaluated using the NRS (Numeric Rating Scale) score [17], while neuropathic pain was assessed using the DN4 (Neuropathic Pain 4 Questions) survey [18,19]. The SF-36 (Short Form Health Survey 36) survey was used to evaluate the quality of life [20].

Once the diagnosis of cervicobrachial pain was made or confirmed, patients were evaluated for appropriate prescriptions considering the Body Mass Index, age, gender, and comorbidities. Only patients whose drug treatments were not effective were scheduled for a cycle of O_2_–O_3_ cervical intramuscular infiltrative therapy, adding on to their current drug therapies. The follow-ups were performed after 8 (T1) and 12 (T2) sessions of topical O_2_–O_3_ therapy. Each patient was monitored using telemedicine, and patients with a new instance of brachial pain were immediately admitted to the hospital. We added only the data after 1 year because, until this period, we did not record any impairments due to symptoms. The development of adverse drug reactions (ADRs) during the treatment with O_2_–O_3_ was evaluated in agreement with our previous studies [21,22].

### 2.3. Local Infiltration

Subject to written informed consent, patients underwent cervical intramuscular injections of an O_2_–O_3_ mixture with an O_3_ concentration of 10 mcg/mL in the cervicobrachial tract (5 mL for each site of injection, usually in the paravertebral site in the region of the brachial plex and the median nerve, bilaterally, for a total of 30 mL) [23].

The injections were performed bilaterally to improve the oxygenation in the body region, bis a week for the first month and once a week for the second month. After the injections, the area was massaged with a topical compound to increase the activity of O_2_–O_3_ in the tissues. The gas mixture was obtained using an ozone generator (Ozo2 Alnitec, Cremosano (CR), Italy) connected to a pure O_2_ source.

### 2.4. Inclusion and Exclusion Criteria

We enrolled patients aged >18 years with clinical symptoms of cervicobrachial pain. Patients previously treated with ozone or patients with blood diseases (i.e., hemolytic anemia or glucose-6-phosphate dehydrogenase deficiency), pregnancy (a relative contraindication), uncontrolled hyperthyroidism, severe cardiovascular diseases, and heart failure were excluded.

### 2.5. Endpoints

The primary clinical endpoint was the statistically significant reduction in pain at T1 (end of the study) compared with T0 (admission) in terms of changes in NRS.

The secondary clinical endpoint was a significant change in the DN4 score and/or in the SF36 value for T2 vs. T1 vs. T0.

Finally, the primary safety endpoint was considered to be the absence of ADRs or drug–drug interactions (DDIs) related to ozone administration. The persistence of symptoms in T2 vs. T1 and/or the development of ADRs that could lead to the discontinuation of treatment were defined as clinical failures.

### 2.6. Statistical Analyses

Data are presented as mean ± standard deviation (SD). At baseline, the independent sample 2-tailed *t*-test was used to compare variables. For categorical parameters, the chi-square test was used. Changes from baseline to end of therapy were analyzed using ranked one-way analysis of variance (ANOVA) with a term for the treatment group. Both Kruskal–Wallis and sign tests were used for non-parametric variables. The Shapiro–Wilk test was used to evaluate the normality of distribution. To reduce the error related to false positives, our hypotheses were tested using Bonferroni-adjusted alpha levels of 0.025 per test. *p*-values < 0.05 were considered statistically significant. Statistical analysis was performed using SPSS 21.0 (International Business Machines Corporation, Armonk, NY, USA). To evaluate the effects of sex and age on both the efficacy and safety of the intramuscular administration of O_2_–O_3_, we performed a sub-analysis of the data.

## 3. Results

### 3.1. Patients’ Basic Characteristics

During the study, 828 patients were eligible. In total, 540 patients (65.2%) 34–87 years old (mean age 61.2 ± 11.7 years) were enrolled and completed the study (Figure 1). In particular, we enrolled 240 men and 300 women with a mean age of 62.6 ± 10.5 years and 59.2 ± 12.3 years, respectively (*p* = 0.2) The most common comorbidities were obesity and blood hypertension (37.8%), followed by type 2 diabetes (24.4%), with differences relating to both sex (Table 1) and age (Table 2).

### 3.2. Pain and Quality of Life 

At admission (T0), an NRS questionnaire documented a severe pain level of 7.7 ± 1.3 (men: 7.75 ± 1.25; women: 7.72 ± 1.37; *p* = 0.939) without correlation with respect to age (men: r 0.079925; women: r −0.00887) or BMI (men: r 0.302446; women: r −0.10572); in 110 patients (20.4%), we documented neuropathic pain (DN-4: 5.8 ± 3.9) (Table 3); in 70 patients (13%), we documented nociplastic pain; and in 360 patients (66.7%), we documented a nociceptive pain. In all enrolled patients, an SF-36 questionnaire documented a low level of quality of life (Table 4). All patients documented a chronic use of at least one drug (range 1–5; mean 1.9 ± 1.06). The most common drugs used were acetaminophen and n-acetyl carnitine (Figure 2). All the patients received a mean of 11.7± 3 cycles of treatment (men: 12.5 ± 1.6; women: 11.1 ± 3.1; *p* = 0.107).

### 3.3. O_2_–O_3_ Treatment

As reported in Figure 1, all patients completed the treatment protocol. At T1, O_2_–O_3_ administration induced a statistically significant (*p* < 0.01) improvement in quality of life, recorded using an SF-36 score (Table 4), and this effect was maintained at the follow-up (T2) (Table 4). At T1, we recorded a statistically significant decrease (*p* < 0.01) in both the DN4 (T1: 4.04 ± 4.37) (Table 3) and NRS values (1.3 ± 1.5). An NRS questionnaire documented a mild pain of 1.3 ± 1.5 (men: 1 ± 1.26; women: 1.52 ± 1.64; *p* = 0.234543) without correlation with respect to age (men: r 0.302446; women: r 0.033841) or BMI (men: r −0.29639; women: r 0.122636). These effects were also maintained during the entire study until the follow-up (T2) (Table 3).

Patients subjected to the O_2_–O_3_ treatment significantly reduced (*p* < 0.01) the use of drugs considering both sex (men: 0.65 ± 0.81; women: 0.72 ± 0.74) (Table 5) and age (Figure 3). 

Finally, during the study, we did not record the development of serious adverse drug reactions able to stop the treatment or the use of other drugs.

## 4. Discussion

In this study, we evaluated the efficacy and safety of topical O_2_–O_3_ therapy in patients with cervicobrachial pain. Cervical pain is commonly due to cervical spondylolysis or osteoarthritis, a chronic degenerative condition inducing changes in bones, intervertebral discs, and/or joints connected to the neck. 

We documented that O_2_–O_3_ therapy reduced pain and DN4, improving the quality of life in both sexes without differences considering age. 

The clinical efficacy of O_2_–O_3_ therapy may be related to the mechanism of action of this compound. In fact, O_2_–O_3_ can elicit the upregulation of antioxidant enzymes such as superoxide dismutase (SOD), GSH-peroxidases (GSH-Px), GSH-reductase (GSH-Rd), and catalase (CAT), inducing an antioxidant response able to reduce the chronic oxidative stress [24]. 

Moreover, O_2_–O_3_ decreases the NF-kB pathway and inhibits the cascade of proinflammatory cytokines involved in the chronic inflammatory process and in pain [25]. 

Several studies highlight the efficacy of O_2_–O_3_ therapy in cervicobrachial pain; Alexandre et al. [26] assessed the efficacy of a single intradiscal injection of O_2_–O_3_ preceded and followed by 5 intramuscular paravertebral injections in the treatment of 252 patients with cervical disc herniation, documenting a significant decrease in pain in 79.3% and sensory dysfunction in 78.1%. In 61.9% of patients, the authors showed the complete regression of motor deficits.

Raeissadat et al. [27] documented that O_2_–O_3_ and lidocaine treatments showed superior, although not statistically different, results compared with a dry needling group. In a case series, Martinelli et al. [16] studied the safety and effectiveness of intramuscular–paravertebral injections of O_2_–O_3_ (O_3_ concentration of 16 mcg/mL once a week) in 168 patients affected by cervicobrachial pain, showing a significant pain reduction (*p* < 0.001) at follow-ups after 1, 2, 3, 4, and 5 years. 

Beyaz et al. [28] investigated the 6-month efficacy and safety of intradiscal O_2_–O_3_ mixture therapy in 44 patients with cervical disc herniation and chronic neck pain. A 73.1% decrease in the average VAS score compared with the baseline values at the final follow-up was observed. In total, 88.6% of patients were satisfied, 9.1% were moderately satisfied, and 2.3% were poorly satisfied. 

No data related to the effect of this treatment on drug use have been published. In the present study, we documented that all enrolled patients received a drug treatment to reduce pain. These treatments did not reduce the clinical symptoms. In contrast, intramuscular treatment with O_2_–O_3_ induced a time-dependent decrease in pain with an improvement in quality of life, as recorded using the SF-36 scale. Moreover, this treatment significantly reduced the use of drugs in both sexes and all ages. This point is very relevant because, in elderly patients, the high number of drugs increases the risk of drug interaction with a decrease in quality of life [29,30].

Low O_2_–O_3_ efficacy could be related to (i) an imprecise ozone generator; (ii) an imprecise gas volume; (iii) an undefined ozone concentration; or (iv) a nonoptimal dose for achieving a therapeutic effect. 

Before the administration of O_2_–O_3_, the concentration should be set to a specific range to ensure safety; after injections, patients might feel a little burning and/or a sensation of heaviness at the injection site that spontaneously decreases in a few minutes. Adverse effects might be related to an incorrect administration technique, including pain, hematoma, infections in the injection site, vagal crisis, and even death [31]. In our study, among the enrolled patients, none had ADR even if they experienced a transient burning in the infiltration sites and redness, which disappeared after a few minutes. 

In agreement with the statistical analyses, we can argue that, in neck and brachial pain, the intramuscular administration of O_2_–O_3_, as an add-on to pharmacological treatments, can favor an improvement in the clinical conditions of patients.

In our study, we documented that before the treatment the patients reported the presence of pain and fatigue with a decrease in physical functioning that induced, particularly in women, a decrease in social functioning with the presence of anxiety and depression. After the O_2_–O_3_ treatment, we documented a statistically significant reduction in the pain rating scales and a statistically significant improvement in quality of life. Subsequent studies are required to broaden the patient sample and to evaluate the efficacy of placebo-controlled oxygen–ozone infiltrative therapy alone. The present study has some limitations. The most important is the absence of a control group; it is impossible to use an intramuscular treatment with a placebo (it is not ethical) or other drugs (e.g., corticosteroids or anesthetics, poorer appropriate drugs). 

In conclusion, we reported that intramuscular–paravertebral O_2_–O_3_ injections represent an effective, safe, conservative approach in patients affected by cervical pain, particularly in patients with comorbidities and polytherapy.

## Figures and Tables

**Figure 1 jcm-12-00248-f001:**
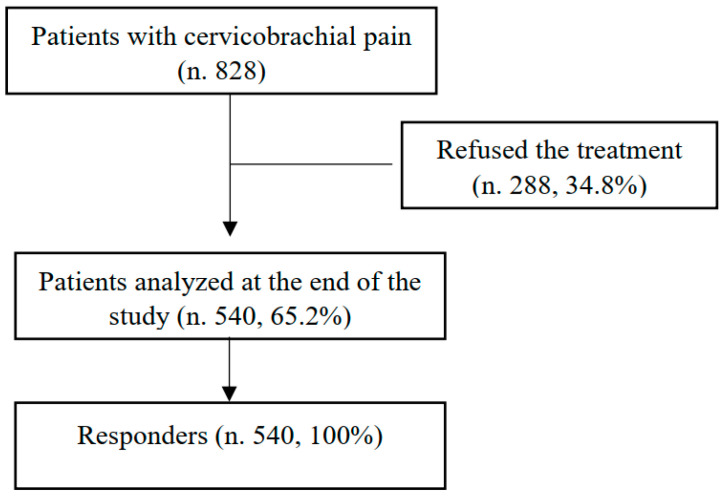
Patient selection flow diagram.

**Figure 2 jcm-12-00248-f002:**
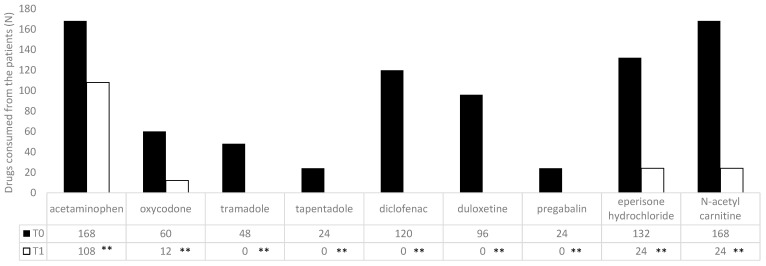
Drugs most commonly used by the patients at the time of enrollment (T0) and after the treatment (T1). Data are expressed as the absolute value. ** *p* < 0.01, T1 vs. T0.

**Figure 3 jcm-12-00248-f003:**
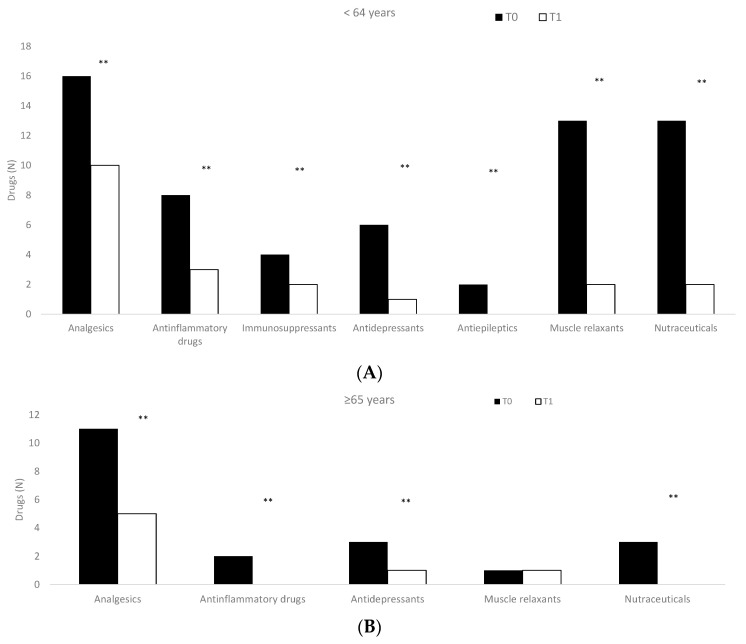
Drugs used in enrolled patients considering age ((**A**): <64 years; (**B**): ≥65 years) at the time of enrollment (T0) and after the treatments (T1). ** *p* < 0.01, T0 vs. T1.

**Table 1 jcm-12-00248-t001:** Demographic characteristics of enrolled patients (n. 540) at the time of admission (T0). Data are expressed as absolute values with respect to the sex evaluation.

	Men (N: 240)	%	Women (N: 300)	%	Delta Percentage of Men vs. Women (%)	*p*
Years						
18–64	132	55	216	72	−17.00	<0.05
>65	108	45	84	28	17.00	<0.05
BMI kg/m^2^						
<24	84	35	84	28	7.00	
25–30	84	35	84	28	7.00	
>30	72	30	132	44	−14	<0.05
Comorbidity						
Anxiety/depression	0	0	60	20	−20.0	<0.01
Arthritis	0	0	36	12	−12.0	<0.05
Benign prostatic hyperplasia	26	10.8	0	0	10.8	<0.05
Blood hypertension	96	40	108	36	4.0	
Bowel inflammatory diseases	12	5	0	0	5.0	
Chronic obstructive pulmonary disease	12	5	24	8	−3.0	
Gastroesophageal reflux diseases	36	15	12	4	11.0	<0.05
Heart disease	24	10	12	4	6.0	
Hypercholesterolemia	36	15	84	28	−13.0	<0.05
Obesity	72	30	84	28	2.0	
Obstructive sleep apnea	12	5	24	8	−3.0	
Osteoarthritis	72	30	0	0	30.0	<0.01
Osteoporosis	0	0	12	4	−4.0	
Sjogren syndrome	0	0	24	8	−8.0	
Thyroiditis	0	0	48	16	−16.0	<0.05
Type 2 diabetes	96	40	36	12	28.0	<0.01
Drug use						
Analgesics	168	70	192	64	6.0	
Anti-inflammatory drugs	63	25	108	36	−9.8	
Immunosuppressants	12	5	24	8	−3.0	
Antidepressants	48	20	60	20	0.0	
Antiepileptics	12	5	12	4	1.0	
Muscle relaxants	64	25	108	36	−9.3	
Nutraceuticals	36	15	156	52	−37.0	<0.01

*p*-values are reported only for significant values. BMI: Body Mass Index.

**Table 2 jcm-12-00248-t002:** Demographic characteristics of enrolled patients (n. 540) evaluated considering the age (<65 and ≥65) at the time of admission (T0). Data are expressed as absolute values respect to the age evaluation.

	<65 Years (N: 348)	%	≥65 Years (N: 192)	%	Delta Percentage of Young vs. Old (%)	*p*
Men	132	37.9	108	56.25	−18.3	0.02
Women	216	62.1	84	43.75	18.3	0.02
Mean age	54.3 ± 7.2		73.6 ± 7.6			
BMI kg/m^2^						
<24	60	17.2	36	18.75	−1.5	
25–30	36	10.3	60	31.25	−20.9	0.009
>30	12	3.45	96	50	−46.6	0.001
Comorbidity						
Anxiety/depression	60	17.2	0	0	17.24	0.03
Arthritis	36	10.3	0	0	10.34	
Benign prostatic hyperplasia	12	3.45	12	6.25	−2.80	
Blood hypertension	120	34.5	84	43.75	−9.27	
Bowel inflammatory diseases	0	0	12	6.25	−6.25	
Chronic obstructive pulmonary disease	24	6.9	12	6.25	0.65	
Gastroesophageal reflux diseases	36	10.3	36	18.75	−8.41	
Heart disease	0	0	12	6.25	−6.25	
Hypercholesterolemia	84	24.1	36	18.75	5.39	
Obesity	108	31	96	50	−18.97	0.02
Obstructive sleep apnea	36	10.3	0	0	10.34	
Osteoarthritis	24	6.9	48	25	−18.10	0.02
Osteoporosis	12	3.45	0	0	3.45	
Sjogren syndrome	24	6.9	0	0	6.90	
Thyroiditis	24	6.9	24	12.5	−5.60	
Type 2 diabetes	24	6.9	108	56.25	−49.35	0.001
Drug use						
Analgesics	192	55.2	168	87.5	−32.33	0.009
Anti-inflammatory drugs	111	31.9	60	31.25	0.65	
Immunosuppressants	36	10.3	0	0	10.34	0.049
Antidepressants	72	20.7	36	18.75	1.94	
Antiepileptics	24	6.9	0	0	6.90	
Muscle relaxants	156	44.8	15	7.8125	37.02	0.001
Nutraceuticals	156	44.8	36	18.75	26.08	0.009

*p* values were reported only for significant values. BMI: Body Mass Index.

**Table 3 jcm-12-00248-t003:** DN-4 scores. Data represent the mean ± standard deviation of the score achieved at admission (T0), at the end of the study (T1), and at the follow-up (T2).

Item	T0	T1	T2
Burning	0.76 ± 0.43	0.62 ± 0.49 **	0.62 ± 0.49
Painful cold	0.70 ± 0.46	0.59 ± 0.50 **	0.59 ± 0.50
Electric shocks	0.19 ± 0.40	0.16 ± 0.37 **	0.16 ± 0.37
Pain and symptoms			
Tingling	0.83 ± 0.38	0.65 ± 0.48 **	0.65 ± 0.48
Pins and needles	0.05 ± 0.23	0.05 ± 0.23 **	0.05 ± 0.23
Numbness	0.89 ± 0.31	0.54 ± 0.51 **	0.54 ± 0.51
Itching	0.65 ± 0.48	0.19 ± 0.40 **	0.21 ± 0.43
Pain located in an area			
Hypoesthesia to touch	0.62 ± 0.49	0.46 ± 0.51 **	0.46 ± 0.51
Hypoesthesia to prick	0.22 ± 0.42	0.19 ± 0.40 **	0.19 ± 0.40
Pain caused or increased			
Brushing	0.89 ± 0.31	0.59 ± 0.50 **	0.59 ± 0.50

** *p* < 0.01, T1 vs. T0 and T2 vs. T1.

**Table 4 jcm-12-00248-t004:** SF-36 scores representing the percentage of the total possible scores achieved at admission (T0), at the end of the study (T1), and at the follow-up (T2).

	T0	T1	T2
Physical functioning	65 ± 18.6	92 ± 7.5 **	91 ± 8.8
Limitations due to physical health	2.1 ± 1.8	75.7 ± 14.3 **	73 ± 15.6
Limitations due to emotional problems	1.2 ± 0.9	97.6 ± 4.3 **	96.9 ± 4.9
Energy/fatigue	45 ± 8.2	65.3 ± 13.2 **	63.4 ± 14.5
Emotional well-being	68 ± 9.7	68.1 ± 11.3 **	66.7 ± 11.4
Social functioning	37.3 ± 3.8	75.2 ± 16.5 **	73.6 ± 15.2
Pain	22.7 ± 3.2	77.5 ± 12.4 **	72.1 ± 16.5
General health	35.1 ± 5.2	66.2 ± 12.4 **	64.3 ± 10.3
Health change	25.3 ± 6.1	92.8 ± 6.5 **	91.6 ± 7.1

Data are expressed as mean ± standard deviation. ** *p* < 0.01, T1 vs. T0 and T2 vs. T1.

**Table 5 jcm-12-00248-t005:** Drug use at the time of admission (T0) and after the O_2_–O_3_ cycle of treatment (T1 and T2) in enrolled patients (N. 540). Data are expressed as the absolute value. Data are expressed as the mean ± standard deviation.

	Men			Women		
	T0	T1	T2	T0	T1	T2
Analgesics	168	108 **	108	192	96 **	96
Anti-inflammatory drugs	60	0 **	12 **	108	36 **	48 **
Immunosuppressants	12	12	12	24	12 **	12
Antidepressants	48	12 **	12	60	24 **	24
Antiepileptics	12	0 **	0	12	0 **	0
Muscle relaxants	60	12 **	12	108	24 **	24
Nutrients	36	0 **	0	156	24 **	24

** *p* < 0.01, T1 vs. T2 and T2 vs. T1.

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
