# Peer review of "Oxygen–Ozone Therapy in Cervicobrachial Pain: A Real-Life Experience"

_jcm, 2022, doi:10.3390/jcm12010248_

Round 1

Reviewer 1 Report

The study aimed to assess whether intramuscular injections of O2O3 mixture are effective and safe in reducing pain and enhancing the quality of life in patients with cervicobrachial pain.  Overall, the article's topic is important, however, the manuscript presents some major issues that authors should consider. Here are some comments and suggestions.

Abstract:

There is an inconsistency in the number of subjects involved in the study; line 19 “we enrolled 48 subjects”, line 21 “All subjects (n=540)”. Please clarify.

The study is defined as “observational”, however, a treatment was administered to the participants (local infiltration). Please clarify.

Introduction

Lines 56-57: The aim of the study is vague. Although well detailed in the Methods section, I suggest better explaining the main scope of the study in the introduction.

Methods

Line 60, the research is described as a pilot study, however in the abstract is described as a prospective, observational, study. Please use a consistent definition of your study design throughout the manuscript.

Line 96 “The primary clinical endpoint was the statistically significant improvement” I would replace the word improvement with reduction.

Statistical analysis

Data are summarized through their means and standard deviation. Was the normality of distribution checked? I suggest reporting how the normality of distribution was checked in the paragraph to legitimate the use of means and SDs to resume the data. The same applies to the use of parametric and non-parametric tests.

When ANOVA or Kruskal Wallis tests were performed did the authors apply any correction for pairwise comparisons (e.g. Bonferroni)? I suggest applying a correction or if already applied, specifying it in the text.

Results

I believe it would be relevant for the reader to know whether there were differences in baseline characteristics between those who agreed to receive the treatment and those who did not. If available I would add this comparison to the supplementary materials.

It is interesting the comparison between women and men and between old and young participants. However, this comparison is not mentioned among the scopes of the study. Please explain the rationale behind those comparisons in the Methods section. How was the cut-off of 65 years chosen?

Table 1: BMI is usually reported as Body Mass Index rather than Body Max Index.

Table 1: Why in drug use only one p-value is reported? The same applies to comorbidities and Table 2. Were those not reported above 0.05? If the case, please report the p-value anyway. I also suggest reporting the p-value for each comparison instead of p<0.05 or p<0.01.

Table 2: please check the sentence in the caption: “Data are expressed as absolute values 125 respect to the sex evaluation”

Table 2: delta percentage reported is between subjects aged <65 and >=65 or between men and women as written?

Figure 2: the title of the y-axis is drugs, if I have correctly understood the figure, should it be the number of patients taking that drug instead?

Figure 2: was a comparison made between the number of participants taking each drug at T0 and T1? Although p<0.001 is mentioned in the caption, I don’t see it reported in the figure.

Table 3: were SF-36 scores compared across time points? Although reported in the text, I don’t see it in the table. Please add p-values to Table 3 and asterisks where pairwise comparisons remained significant after correction. The same applies to Table 4.

Since the first endpoint of the study was the reduction of the NRS score over time I would report NRS values recorded at T0, T1, and T2 in a table, maybe together with the SF-36 score, to save space. Similarly, I suggest also reporting the DN4 score recorded at each time point, which is the second endpoint of the study.

Figure 3: it is not clear what p<0.01 refers to nor the line over the bars. Please clarify and modify the figure accordingly.

Discussion

The authors correctly state that further studies including a control group are required. Why the presence of a control group was not considered in the design of this study?

Please add a limitations section to the Discussion paragraph.

Reviewer 2 Report

The aim of this study was to carry out the clinical efficacy and safety of intramuscular paravertebral injections of an oxygen-ozone (O2-O3) mixture in patients with cervicobrachial pain.

My opinion is that researches analyzing the effect of oxygen-ozone (O2-O3) therapy are needed because it is a current strategy nowadays. Moreover, I believe that due to the good results there will be a growth in its use in the future. Nevertheless, the study should be revised due to some minor spelling mistakes and the fact that the authors did not follow the guidelines used by the journal.

In the introduction, the mechanism of oxygen-ozone therapy should be explained in a detailed way and given some information in relation to the present knowledge in which concerns the effect of this intervention in spine disorders, namely if results of other studies are consensual or not, and the reasons to have some contradictory results.

Methods:

The injection was done “in the cervicobrachial tract (5 mL for each site of injection bilaterally, for a total of 30 mL)”. In this sense, the information of the exact point of the cervicobrachial region where the injection is applied is not clearly expressed. In addition, the reason why the injection for the treatment is done bilaterally.

Were the following factors (origin of pain, the duration of symptoms, the intensity of symptoms, the age of patients, and emotional characteristics of patients) taken into account when selecting the sample? If not, the idea can be misleading because it seems that this kind of treatment can be used in patients with different kinds of symptoms. 

The dose of treatment was the same for every patient. It would be relevant to discuss whether a personal treatment should be considered having in mind some characteristics such as age, BMI, symptoms, gender.

Although in the abstract you mention that the patients were free of pain after a year, there is no such data throughout the article. Wasn´t there any follow up appointment after the last one (12 sessions) to see if the treatment was still effective in the long run? If not, it should be a strong limitation. 

In the discussion, it should be interesting to understand why there was such a high percentage of patients refusing this intervention?

In the end of the discussion, it is said that there was significant improvement:

“There is a statistically significant improvement in the pain rating scales and in the

quality of life.”  However, you then say: “Patients reported the presence of pain, the decrease in physical functioning and the presence of fatigue. These induced in women a decrease in social functioning with presence of anxiety and depression.” Isn’t this a contradiction? 

Author Response

Dear Reviewer, thank you for your comments, we have revised the manuscript considering your suggestion and now we send you the last version of the manuscript with the responses point by point to your questions.

The aim of this study was to carry out the clinical efficacy and safety of intramuscular paravertebral injections of an oxygen-ozone (O2-O3) mixture in patients with cervicobrachial pain.

My opinion is that researches analyzing the effect of oxygen-ozone (O2-O3) therapy are needed because it is a current strategy nowadays. Moreover, I believe that due to the good results there will be a growth in its use in the future. Nevertheless, the study should be revised due to some minor spelling mistakes and the fact that the authors did not follow the guidelines used by the journal.

In the introduction, the mechanism of oxygen-ozone therapy should be explained in a detailed way and given some information in relation to the present knowledge in which concerns the effect of this intervention in spine disorders, namely if results of other studies are consensual or not, and the reasons to have some contradictory results.

R: in agreement with your suggestions, we added more data concerning the mechanism of action of oxygen-ozone therapy in introduction and in discussion

Methods:

The injection was done “in the cervicobrachial tract (5 mL for each site of injection bilaterally, for a total of 30 mL)”. In this sense, the information of the exact point of the cervicobrachial region where the injection is applied is not clearly expressed. In addition, the reason why the injection for the treatment is done bilaterally.

R: in agreement with your suggestions, we added these informations.

Were the following factors (origin of pain, the duration of symptoms, the intensity of symptoms, the age of patients, and emotional characteristics of patients) taken into account when selecting the sample? If not, the idea can be misleading because it seems that this kind of treatment can be used in patients with different kinds of symptoms. 

R: yes we used this treatment only in patients without improvement of symptoms during the drug treatment, as you can read the intensity of the pain was strong in all the enrolled patients.

The dose of treatment was the same for every patient. It would be relevant to discuss whether a personal treatment should be considered having in mind some characteristics such as age, BMI, symptoms, gender.

R: during the treatment all the patients in the first time were evaluated for appropriate prescription considering the Body Mass Index, the age the gender and the comorbidity. Only patients without efficacy of the drug treatments were treated with O2O3

Although in the abstract you mention that the patients were free of pain after a year, there is no such data throughout the article. Wasn´t there any follow up appointment after the last one (12 sessions) to see if the treatment was still effective in the long run? If not, it should be a strong limitation. 

R: all the patients were evaluated using the telemedicine and patients with a new insurgence of brachial pain were immediately admitted. We added only the data after 1 year because until this period we did not record any impairments of symptoms. However, we clarify this in the text, line 165.

In the discussion, it should be interesting to understand why there was such a high percentage of patients refusing this intervention?

R: this is a real-life study and usually the patients refused a continuous intramuscular treatment for faire of injections and for faire of O2O3.

In the end of the discussion, it is said that there was significant improvement:

“There is a statistically significant improvement in the pain rating scales and in the

quality of life.”  However, you then say: “Patients reported the presence of pain, the decrease in physical functioning and the presence of fatigue. These induced in women a decrease in social functioning with presence of anxiety and depression.” Isn’t this a contradiction? 

R: sorry, there has been an error during the last revision. We revised this point (see 215-224)

Round 2

Reviewer 1 Report

The manuscript has improved a lot after revision. I still suggest some comments for minor revisions.Please find them highlighted in yellow.

Author Response

Dear Reviewer 1

I have read your comments and we performed all changes as you have requested
